# Effect of Combined Bee Venom Acupuncture and NSAID Treatment for Non-Specific Chronic Neck Pain: A Randomized, Assessor-Blinded, Pilot Clinical Trial

**DOI:** 10.3390/toxins13070436

**Published:** 2021-06-23

**Authors:** Boram Lee, Byung-Kwan Seo, O-Jin Kwon, Dae-Jean Jo, Jun-Hwan Lee, Sanghun Lee

**Affiliations:** 1Clinical Medicine Division, Korea Institute of Oriental Medicine, Daejeon 34054, Korea; qhfka9357@kiom.re.kr (B.L.); cheda1334@kiom.re.kr (O.-J.K.); 2Department of Acupuncture & Moxibustion, Kyung Hee University Hospital at Gangdong, Seoul 05278, Korea; seohbk@hanmail.net; 3Department of Neurosurgery, Kyung Hee University Hospital at Gangdong, Seoul 05278, Korea; apuzzo@hanmail.net; 4Korean Medicine Life Science, Campus of Korea Institute of Oriental Medicine, University of Science & Technology (UST), Daejeon 34054, Korea; ezhani@kiom.re.kr; 5Future Medicine Division, Korea Institute of Oriental Medicine, Daejeon 34054, Korea

**Keywords:** bee venom acupuncture, pharmacopuncture, non-specific chronic neck pain, randomized controlled trial

## Abstract

This study aimed to investigate the feasibility of a combined treatment of bee venom acupuncture (BVA) and non-steroidal anti-inflammatory drugs (NSAIDs) for the treatment of non-specific chronic neck pain (NCNP). Patients with NCNP for ≥3 months were randomly allocated to a BVA, NSAIDs, or combined group (1:1:1), receiving 6 sessions of BVA, loxoprofen (180 mg daily), or a combination, respectively, for 3 weeks. Recruitment, adherence, and completion rates were calculated to assess feasibility. Bothersomeness, pain, disability, quality of life, depressive status, treatment credibility, and adverse events were assessed. In total, 60 participants were enrolled, and 54 completed the trial. Recruitment, adherence, and completion rates were 100%, 95%, and 90%, respectively. Bothersomeness, pain, disability, and depressive symptoms significantly improved in all groups after treatment (*p* < 0.05). The combined group showed continuous improvement during the follow-up period (*p* < 0.05). Quality of life was significantly improved (*p* < 0.05), and treatment credibility was maintained in the BVA and combined groups. No serious adverse events were reported. Combined treatment of BVA and NSAIDs are feasible for the treatment of NCNP, showing high persistence of the effect, credibility, and safety. Additional trials with longer follow-up are needed to confirm this effect.

## 1. Introduction

Non-specific chronic neck pain (NCNP) refers to persistent pain or severe discomfort in the neck for at least 3 months, without pathological and neurological findings and traumatic causes [1,2]. The lifetime prevalence of neck pain worldwide is estimated as 14–71% [3]. It is higher in women, office and computer workers, high-income countries, and urban areas [4]. The global burden of neck pain expressed in disability-adjusted life years increased from 23.9 million in 1990 to 33.6 million in 2010, causing a high socioeconomic burden, such as loss of productivity and increased absenteeism, as well as disability [5].

Conventional treatments for NCNP include pharmacotherapy, physical therapy, and surgical interventions. Non-steroidal anti-inflammatory drugs (NSAIDs) are commonly used for the treatment of NCNP due to anti-inflammatory and analgesic effects [6]. However, a lack of long-term efficacy and possible risk of gastrointestinal, kidney, and cardiovascular side effects [7] has led to the use of complementary and integrative therapies, including acupuncture and massage therapy [8,9].

Pharmacopuncture is an acupuncture technique that injects herbal medicine into acupoints, inducing the effects of both herbal medicine and acupuncture. Bee venom acupuncture (BVA) is a representative pharmacopuncture that uses diluted bee venom, exerting both mechanical function from acupoint stimulation and pharmacological action from the bioactive compounds isolated from bee venom. Bee venom induces allergic immune responses such as increased CD16 surface expression [10]. However, its repeated administration can induce therapeutic effects such as anti-nociceptive effects by desensitization [11,12]. Therefore, BVA has been widely used for the treatment of musculoskeletal diseases, including neck pain [13,14,15]. Additionally, it has improved efficacy when compared with acupoint stimulation alone and non-acupoint bee venom injection [16,17]. Many reports have shown the analgesic, anti-inflammatory, anti-arthritis, and anti-cancer effects of BVA in experimental models [11,16,18]. In addition, the clinical effect of BVA on NCNP has been reported [13,19].

As the importance of multimodal approach in pain management is being emphasized [20,21], combinations of conventional treatment, such as NSAIDs, and complementary and integrative therapies, such as BVA, have been utilized in real world practice. However, to the best of our knowledge, there has been no report evaluating the efficacy of the combined treatment of NSAIDs and BVA compared with single therapies. The objective of this pilot study was to investigate the feasibility of the combined treatment of BVA and NSAIDs in patients with NCNP.

## 2. Results

### 2.1. Study Participants and Feasibility Outcomes

A total of 62 participants were screened for eligibility, of which 60 were randomly assigned to the BVA (n = 20), NSAIDs (n = 20), and combined treatment (n = 20) groups (recruitment rate, 96.8%). Of these, 54 completed the study, and 6 dropped out (1 in the BVA, 3 in the NSAIDs, and 2 in the combined group). Two participants (1 in the BVA group and 1 in the combined group) withdrew their consent to participate in the trial because of personal reasons not related to the BVA and four participants (3 in the NSAIDs group and 1 in the combined group) violated the study protocol by taking prohibited medications, including steroid injection and acupuncture (Figure 1) (Appendix A).

The BVA adherence rates in the BVA and combined groups were each 100% (20/20) and 90% (18/20), respectively (overall, 95% [38/40]), and the completion rates of the BVA, NSAIDs, and combined groups were 95% (19/20), 85% (17/20), and 90% (18/20), respectively (overall, 90% [54/60]). In the BVA and combined groups, compliance with BVA treatment for the corresponding clinical trial period was 100%. In addition, mean compliance rate with NSAIDs in the NSAIDs and combined groups was 87.10% (95% confidence interval (CI) 82.45, 91.75) and 90.95% (95% CI 86.27, 95.63), respectively (overall mean, 89.03%; 95% CI, 85.82, 92.23).

There were no significant differences in the baseline demographic characteristics between the groups (*p* > 0.05; Fisher’s exact test (sex, smoke, and drink) and one-way analysis of variance (ANOVA) (height, weight, body mass index, and vital signs)), except age (*p* < 0.0001; one-way ANOVA). Post hoc analysis, according to Scheffé’s method, showed that the age of the NSAIDs group was significantly lower than the other groups; therefore, age was used as a covariate for the outcome analysis to minimize potential bias. Additionally, the proportion of women was higher than that of men in the study population. Body mass index was in the normal or overweight range for most patients, and most did not smoke or drink alcohol (Table 1).

### 2.2. Primary Outcome

In the mean visual analogue scale (VAS) for bothersomeness, there was significant reduction of 34.60 (95% CI 26.46, 42.74), 24.55 (95% CI 14.02, 35.08), and 22.25 (95% CI 13.64, 30.86) at Week 4 (our primary time point) compared with baseline in each group (all, *p* < 0.0001). The difference between the three groups was significant (*p* = 0.0317). Post hoc analysis, according to Scheffé’s method, showed that the VAS for bothersomeness significantly improved in the BVA group when compared with the combined group (mean difference (MD) −16.30, 95% CI −31.48, −1.11). At Week 8 follow-up, the BVA group maintained a decreased state of VAS for bothersomeness, and the NSAIDs group slightly increased, while VAS in the combined group continued to decrease after the treatment ended. The difference between the three groups was significant (*p* = 0.0214). Post hoc analysis revealed that the BVA group was significantly improved when compared with the NSAIDs group (MD −20.93, 95% CI −39.37, −2.49). During the clinical trial period, bothersomeness significantly decreased in all groups compared with before treatment, and the repeated measured analysis of variance (RM-ANOVA) results showed that the interaction effect on time and treatment group was significant (*p* = 0.0331; Figure 2 and Table 2).

### 2.3. Secondary Outcomes

At Week 4, all groups significantly improved in VAS for pain intensity (*p* < 0.0001 in the BVA and combined groups, *p* = 0.0011 in the NSAIDs group) and neck disability index (NDI) score (*p* < 0.0001, *p* = 0.0196, and *p =* 0.0039 in the BVA, NSAIDs, and combined groups, respectively) compared with baseline. Interestingly, the VAS for pain intensity and NDI score continued to decrease at Week 8 follow-up in the BVA (*p* < 0.0001, all) and combined groups (*p* = 0.0002 and *p* = 0.0008). Post hoc analysis showed that there was a significant difference between BVA and combined groups at Week 4 in VAS for pain intensity (MD −14.42, 95% CI −28.76, −0.08) and between BVA and NSAIDs groups in NDI score at Week 8 (MD −11.7, 95% CI −21.1, −2.3). In addition, there was a borderline significant difference between NSAIDs and combined group in favor of the combined group in NDI score at Week 8 (MD 7.5, 95% CI −0.9, 15.9; Figure 2 and Table 2).

As for the health-related quality of life, the EuroQol 5-dimension (EQ-5D) score significantly increased in the BVA and combined groups compared with before treatment during the trial period, although the difference between groups was not significant. The 36-item Short Form Health Survey (SF-36) scores were significantly improved in the BVA group at Week 4 (*p* = 0.0012) and at Week 8 in the BVA and combined groups (*p* = 0.0002 and *p* = 0.0245), compared with baseline. There was significant difference between groups at Week 8 (*p* = 0.0182); post hoc analysis showed a significant difference between BVA and NSAIDs groups (MD 9.44, 95% CI 1.00, 17.89) and a borderline difference between NSAIDs and combined groups (MD −6.86, 95% CI −14.41, 0.68). At Week 4, depressive mood, as measured by the Beck Depression Inventory (BDI), significantly improved in all groups (*p* = 0.0025, *p* = 0.0417, and *p = 0*.0040 in the BVA, NSAIDs, and combined groups, respectively). These significant effects persisted in the BVA (*p* = 0.0001) and combined groups (*p* = 0.0041) at Week 8. There was a significant difference between groups at Week 8 (*p* = 0.0250); post hoc analysis showed a significant difference between the BVA and NSAIDs groups (MD −5.07, 95% CI −9.61, −0.53) and a borderline difference between the NSAIDs and combined groups (MD 7.5, 95% CI −0.9, 15.9; Table 2).

### 2.4. Credibility Test

The credibility test showed no significant differences in the level of confidence in the treatment the participants received between before and after treatment in the BVA and combined groups (all, *p* > 0.05). In the NSAIDs group, credibility decreased significantly at Week 4 compared with baseline (*p* = 0.0308), especially in willingness to recommend to others (*p* = 0.0421) and rationality of treatment (*p* = 0.0153; Table 3).

### 2.5. Safety

Six adverse events occurred in >519 visits during the study period. There was no significant difference in incidence between groups (Fisher’s exact test, *p* = 0.7012). In the BVA and combined groups, two and one case of mild itching and redness at injection sites was reported in >179 (1.12%) and >170 visits (0.59%), respectively. These cases were judged to be definitely related to BVA treatment. In the NSAIDs group, one case each of skin urticaria, abdominal pain and nausea, and low back pain due to lumbar sprain (3 cases/170 visits [1.76%]) was reported. Only abdominal pain and nausea were judged to be definitely related to NSAIDs, and the severity of all adverse events was mild. All adverse events spontaneously subsided, and no serious adverse events occurred during the clinical trial period. In addition, there were no significant changes in vital signs or in laboratory liver and renal function tests before and after treatment in all groups (Appendix A). 

## 3. Discussion

To the best of our knowledge, this is the first randomized controlled trial to assess the feasibility of a combined treatment of BVA and NSAIDs for patients with NCNP. The compliance rate of BVA was very high (100%) in the BVA and combined groups. It was also high for NSAIDs (89.03%) in the NSAIDs and combined groups. The recruitment, adherence, and completion rate were very high (96.8%, 95%, and 90%, respectively) during the clinical period, showing high feasibility. Treatment credibility did not change before and after treatment in the BVA and combined groups. In addition, our results show that 3 weeks of BVA incremental treatment, NSAIDs treatment, or combined treatment significantly improved NCNP bothersomeness. Interestingly, there was no superior effect of combined treatment compared with single treatment; however, bothersomeness continued to decrease at Week 8 follow-up after treatment ended in the combined group alone. This suggests a long-term effect. Pain intensity and disability significantly improved at Week 4 in all groups, and health-related quality of life significantly improved in the BVA and combined groups. In addition, depressive mood significantly improved in all groups at Week 4, and this effect persisted until Week 8 in the BVA and combined groups. Interestingly, the BVA group showed the greatest improvement, and the combined group showed borderline significant differences, compared with the NSAIDs group at Week 8. The compliance rate of NSAIDs was high; therefore, these results could not be interpreted as differences in compliance. In the BVA and combined groups, adverse reactions related to BVA were mild local reactions despite our BVA incremental dose protocol, and no serious adverse reactions occurred.

Interestingly, the treatment effect in the BVA and combined groups persisted for 5 weeks in all outcomes, even after treatment had finished. The NSAIDs group showed only a short-term effect during the 3-week treatment period. The primary outcome, VAS for bothersomeness, showed continuous improvement in the combined group alone at Week 8 follow-up, although the difference was not significant compared with the single treatment groups. These results are similar to a previous study showing that the treatment effect of 2 months of BVA and physiotherapy persisted for 1 year compared with normal saline injection plus physiotherapy in patients with adhesive capsulitis of the shoulder [22]. Therefore, future studies should examine the long-term lasting effect of the combined treatment of BVA and NSAIDs through longer follow-up duration with full-scale randomized controlled trials and registry-based studies.

The NSAID used in this study was loxoprofen, one of the most frequently prescribed anti-pain drugs in East Asian countries [23]. Loxoprofen is a non-selective cyclooxygenase (COX) inhibitor with anti-inflammatory, analgesic, and antipyretic effects and has fewer adverse events compared with other NSAIDs [24]. However, gastrointestinal, renal, and cardiovascular disorders have been reported in previous studies [25,26]. The associated adverse events are known to be dose-dependent; therefore, patients should take the lowest effective dose for the shortest period [27]. For this reason, a safe and long-term effective treatment that can complement the NSAIDs is needed. In East Asia, BVA combined with conventional medication, including NSAIDs, has been used to treat cancer-related pain [28] and low back pain [29], and especially, its clinical effect for NCNP has been well studied [13,19]. Future clinical research on the efficacy of combining BVA with a tapered dose of NSAIDs will be a significant contribution in this field.

In our study, BVA alone showed a significant difference compared with the combined group in VAS for bothersomeness and pain intensity at Week 4. This is different from a previous study that reported a combined treatment of morphine and BVA was most effective compared with single treatment [30]. Morphine exhibits its analgesic effect by mediating opioidergic receptors [30]. By contrast, loxoprofen exhibits its analgesic effect via its anti-inflammatory reaction while inhibiting COX-1 and COX-2 enzymes, which are involved in the production of prostaglandins [26]. Therefore, this different mechanism may account for the differences in BVA treatment effects, leading to different research outcomes. Bee venom is composed of various peptides exhibiting anti-inflammatory effects [11]. The analgesic effect of BVA is mediated by the descending pain inhibitory system, including spinal noradrenergic and serotonergic receptors [31,32,33] and modulating immune responses [34]. The anti-inflammatory effect of BVA is associated with the inhibition of COX-2 expression and production of pro-inflammatory cytokines [35]. No serious adverse events occurred in this study, and there were no significant changes before and after treatment in vital signs or laboratory results; however, a combined treatment has the possibility of potential interactions between the two treatments. Therefore, future research should focus on elucidating the underlying mechanisms to understand their possible interactions. In addition, BVA, if not used properly, can cause systemic adverse reactions, such as anaphylactic reactions [36]; therefore, it should be performed by a qualified professional who has completed relevant formal training.

Our study has the strength of assessing the feasibility of the combined treatment of BVA incremental treatment and NSAIDs for participants with NCNP for the first time, which reflects the real-world practice. However, due to the nature of the research design and interventions, we could not perform blinding of participants and practitioners, which might affect the subjective outcome measures of our study. Nevertheless, we sought to apply a rigorous methodology by minimizing detection bias through blinding of the outcome assessor. Additionally, since there was no placebo group in our study, it is difficult to evaluate the effect of each treatment, and it is uncertain whether symptom relief occurred according to the natural course. In future studies, it will be necessary to evaluate the efficacy of the combined treatment of BVA and NSAIDs compared with appropriate controls.

In conclusion, the current study showed high feasibility of combined treatment of BVA and NSAIDs on people with NCNP, showing high recruitment, adherence, and completion rates and high treatment compliance and credibility without serious adverse events. Additional full-scale confirmatory randomized controlled trials on the combined treatment should be performed, especially to examine the long-term effects.

## 4. Materials and Methods

### 4.1. Study Design and Ethics

This was a 3-arm parallel, randomized, assessor-blinded, clinical trial performed at the spine center of Kyung Hee University Hospital at Gangdong (KHUHGD), Seoul, Republic of Korea from July 2013 to July 2014. Participants who voluntarily signed the informed consent after an explanation of the study and met the eligibility criteria were randomly assigned at a 1:1:1 ratio into the 3 weeks of BVA (2 sessions/week, total 6 sessions), NSAIDs, or combined treatment groups. This study was approved by the institutional review board of KHUHGD (approval number: KHNMC-OH-IRB 2012–2019) before the start and was conducted according to Declaration of Helsinki. The protocol of this trial was registered at *ClinicalTrials.gov* (registration number: NCT01922466) and a detailed study protocol has been published [37].

### 4.2. Sample Size Calculation

To the best of our knowledge, this is the first study to assess the feasibility of a combined treatment of BVA and NSAIDs compared with BVA or NSAIDs treatment alone in patients with NCNP. As a pilot study, a sample size of 20 participants in each group (N = 60), was determined to be adequate for assessing feasibility considering the practical considerations and following discussions with clinical experts and a statistician.

### 4.3. Participants

Participants were recruited using online (hospital website) and offline (bulletin boards and local newspaper) advertisements. Written informed consent was obtained after a detailed explanation of the study characteristics, including study objectives, procedures, and potential benefits and harms. Eligibility was evaluated according to pre-determined inclusion and exclusion criteria (Table 4). In addition, as a screening process, 0.05 mL of 1:20,000 BVA was subcutaneously injected at LI11 acupoints to check for hypersensitivity reactions to BVA in all participants. Local swelling >10 mm in diameter and erythema >20 mm in diameter were considered positive reactions, and these participants were excluded from the trial.

### 4.4. Randomization and Blinding

An independent statistician, who was not involved in clinical trial procedure and evaluation, generated random sequence numbers. A simple random number generation was conducted using the “RAND” command in Excel 2007 software (Microsoft, WA, USA), with an allocation ratio of 1:1:1 to each group. Sealed opaque envelopes were used for allocation concealment and were kept in double-locked cabinets at hospital. The envelopes were sequentially opened by the clinical research coordinator (CRC) at the baseline visit. Due to the nature of the clinical trial design, the BVA practitioners and participants could not be blinded. Therefore, we blinded the outcome assessors and data analysts to participant allocation by not involving them in the treatment procedures to minimize detection bias. In addition, training was performed that included the prohibition of any conversation about the treatment procedure to maintain the blindness of outcome assessors.

### 4.5. Interventions

A baseline visit was scheduled within 2 weeks of screening visit for participants who met all inclusion and exclusion criteria. In the BVA group, BVA was administered perpendicularly to 12 fixed acupoints (bilateral SI12, SI14, BL11, BL12, TE15, and GB21) at a depth of 0.5–1.0 cm (subcutaneously) without needle retention, using 1 mL 26-gauge sterile disposable syringes (Green Cross Co., Yongin, Korea). Participants felt stiffness when injecting the needle and a skin stimulation response. BVA was performed twice a week for 3 weeks (6 sessions in total), with a weekly dose incremental protocol to minimize adverse events and maximize the treatment effect. This was based on clinical experience and previous studies [29,38]: 0.2 mL for each acupoint at Week 1, 0.4 mL at Week 2, and 0.8 mL at Week 3. The detailed treatment regimen was determined based on the acupuncture and moxibustion medicine textbook [39] and the consensus among Korean medicine specialists. Participants were treated while lying in the prone position after skin sterilization. BVA practitioners were prohibited from communication with participants other than that related to BVA treatment. The BVA was prepared as dried bee venom powder (Yoomil Garden, Hwasun, Republic of Korea), diluted (1:20,000) and filtered in normal saline (0.9% NaCl) at the Korean medical pharmacy of KHUHGD. A hundred milliliters of normal saline was added to 5 mg of dried bee venom powder and dissolved well; after filtering with a 0.22 μg filter, 5 mL of each was filled in a sterilized vial. After confirming the suitability by conducting an environmental culture microorganism test, it was refrigerated at 4–6 °C. Treatment was performed by licensed Korean medicine doctors with >3 years of clinical experience and 6 years of regular curriculum of medical college. To ensure the consistency and standardized BVA treatment during the clinical trial, an educational workshop was conducted prior to the start of the study under the supervision of the principal investigator.

The NSAIDs group was prescribed loxoprofen (Loxonin, 60 mg/tab, Dong Wha Pharm Co., Ltd., Seoul, Korea), with a dose of oral administration of 1 tablet at a time, 3 times a day, for 3 weeks. The combined treatment group received both BVA and NSAIDs treatment for 3 weeks at the same time, and the treatment contents were the same as in each group.

During the clinical period, except for NSAIDs provided in this clinical trial in the NSAIDs or combined group, any concurrent treatment for neck pain was prohibited, including surgical intervention, steroid injection, muscle relaxant, psychotropic drugs, narcotics, physical therapy, acupuncture, and moxibustion. If the participants had been administered analgesics before the trial, they were enrolled after a 15-day washout period to prevent the previously administered analgesics from affecting the clinical trial results. Concomitant use of other drugs that did not affect the interpretation of the results of this clinical trial was allowed under the judgment of the investigator, and all drugs administered during the clinical trial were recorded in the case report form.

### 4.6. Outcomes

Demographic characteristics, including sex, age, height, weight, smoking, drinking, and medical history, were collected from all participants at baseline. To examine the feasibility of a confirmatory clinical trial, we calculated the recruitment rate (number of enrolled participants/number of screened participants), completion rate (number of participants who completed the clinical trial/number of enrolled participants), and BVA adherence rate (number of participants who completed at least 4 of 6 BVA treatments/number of enrolled participants). In addition, compliance rate with BVA and NSAIDs for the corresponding clinical trial period was calculated.

The primary outcome was the 100 mm VAS for bothersomeness due to NCNP measured at Week 4 (1 week after 3 weeks of treatment), the primary time point. Participants rated the clinical severity and discomfort on activities of daily life due to NCNP over the past week, from 0 (absence of bothersomeness) to 100 (worst bothersomeness imaginable) [40,41] at screening, baseline, Weeks 2, 3, 4, and 8.

The secondary outcomes were pain intensity, measured by 100 mm VAS; neck-pain related dysfunction, measured by NDI; quality of life, measured by the EQ-5D and SF-36; and depressive status, measured by the BDI. Pain intensity over the past week was evaluated using 0 (absence of pain) to 100 (worst pain imaginable) VAS at screening, baseline, Weeks 2, 3, 4, and 8 [42]. The NDI consists of 10 items, expressed as a percentage (total possible score 100%) with a high score indicating more dysfunction [43]. We used the validated Korean version of the NDI [44]. The EQ-5D consists of 5 questions about mobility, personal care, daily activities, pain/discomfort, and anxiety/depression (higher score means higher quality of life) [45,46]. Both NDI and EQ-5D were assessed at baseline, Weeks 2, 3, 4, and 8.

The SF-36 evaluates general health status and quality of life using 36 questions; a higher score indicates a better health status [47]. The BDI evaluates the degree of depression using 21 questions. Each question is evaluated on a 4-point scale of 0 to 3 [48]. Higher scores indicate greater depression severity. Both SF-36 and BDI were assessed at baseline and Weeks 4 and 8.

Treatment credibility was assessed using the credibility test, which consists of 4 questions regarding the expectation of NCNP improvement, willingness to recommend to others, rationality of treatment, and effectiveness for alleviating other complaints [49]. Each item was evaluated on a 1-to-6-point scale at baseline and Week 4; higher scores indicate greater confidence.

The safety assessment was conducted by monitoring all adverse events that occurred during the clinical trial by medical examination and participants’ self-report, regardless of their relationship with the intervention. In addition, any significant changes before and after intervention were measured via laboratory liver and renal function tests and vital signs. The adverse events that can occur after BVA include skin reactions at the injection sites, such as pruritus, rash, and swelling, and systemic symptoms, such as headache, generalized myalgia, and nausea [36]. If such adverse events occurred during the study period and they were judged to have a causal relationship with the BVA by the Korean medical doctor (based on onset and disappearance time of symptoms, pattern, participant’s usual condition, and past history), they were recorded as a BVA-related adverse event.

### 4.7. Statistical Analysis

Based on the principle of intention-to-treat analysis, all statistical analyses were performed using SAS Version 9.4 (SAS institute, Inc., Cary, NC, USA) by a statistician independent of treatment procedure and evaluation. Normality of data distribution was confirmed by Kolmogorov–Smirnov test and Shapiro–Wilk test, and equal variance was confirmed by Levene’s test. For baseline characteristic of participants, one-way ANOVA was performed for continuous variables, and Fisher’s exact test was performed for categorical variables. A two-sided test with a significance level of 0.05 was performed using an analysis of covariance (ANCOVA) with baseline as the covariate and each group as the fixed factor to test the differences between the three groups. Next, post hoc Scheffé adjustment was performed to compare the differences between two groups. A paired *t*-test was used to analyze the changes in the results before and after treatment within the groups. RM-ANOVA was performed to explore interactions between time and treatment group. Missing values were corrected by applying the last observation carried forward method. Interim analysis was not performed.

## Figures and Tables

**Figure 1 toxins-13-00436-f001:**
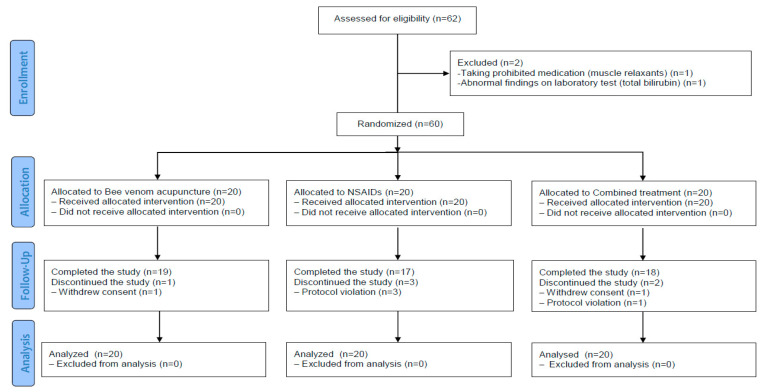
Study flowchart. NSAIDs, non-steroidal anti-inflammatory drugs.

**Figure 2 toxins-13-00436-f002:**
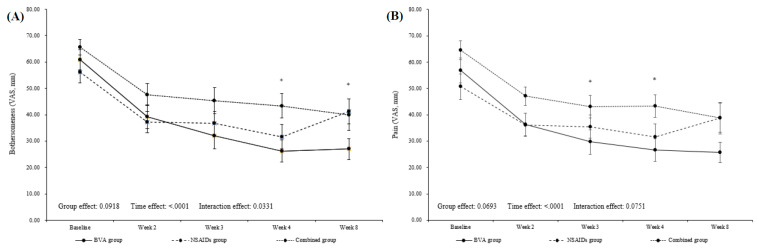
Change over time in visual analogue scale for (**A**) bothersomeness and (**B**) pain. BVA, bee venom acupuncture; NSAIDs, non-steroidal anti-inflammatory drugs; VAS, visual analogue scale. * significant difference between BVA and combined groups, + significant difference between BVA and NSAIDs groups.

**Table 1 toxins-13-00436-t001:** Baseline characteristics of participants.

Characteristics	BVA Group (n = 20)	NSAIDs Group (n = 20)	Combined Group (n = 20)	*p-*Value	*F*-Value
Sex (Male/Female) *	7 (35.0%)/13 (65.0%)	5 (25.0%)/15 (75.0%)	3 (15.0%)/17 (85.0%)	0.4023	-
Age (years) +	52.30 (48.10, 56.50)	37.65 (33.45, 41.85)	45.50 (40.18, 50.82)	<0.0001	11.11
Height (cm) +	164.15 (161.41, 166.89)	164.72 (161.64, 167.80)	161.99 (157.97, 166.01)	0.4447	0.82
Weight (kg) +	63.75 (59.21, 68.28)	62.13 (57.77, 66.49)	59.03 (54.96, 63.09)	0.2674	1.35
BMI (kg/m^2^) +	23.61 (22.18, 25.04)	22.79 (21.76, 23.82)	22.63 (21.44, 23.82)	0.4526	0.80
Smoke (Yes/No) *	2 (10.0%)/18 (90.0%)	2 (10.0%)/18 (90.0%)	1 (5.0%)/19 (95.0%)	0.9999	-
Drink (Yes/No) *	4 (20.0%)/16 (80.0%)	6 (30.0%)/14 (70.0%)	7 (35.0%)/13 (65.0%)	0.6752	-
SBP (mmHg) +	119.8 (113.5, 126.0)	120.1 (112.7, 127.4)	119.6 (112.5, 126.6)	0.9942	0.01
DBP (mmHg) +	78.40 (74.56, 82.24)	79.95 (76.46, 83.44)	74.90 (70.62, 79.18)	0.7290	0.32
Pulse (times/min) +	73.05 (68.91, 77.19)	74.95 (70.86, 79.04)	72.65 (67.30, 78.00)	0.1521	1.95
Temperature (°C) +	36.52 (36.48, 36.56)	36.50 (36.45, 36.55)	36.46 (36.41, 36.50)	0.1147	2.25

BMI, body mass index; BVA, bee venom acupuncture; DBP, diastolic blood pressure; NSAIDs, non-steroidal anti-inflammatory drugs; SBP, systolic blood pressure. * Statistical analysis was performed using Fisher’s exact test, and data are presented as number (%). + Statistical analysis was performed using one-way analysis of variance, and data are presented as means (95% confidence interval).

**Table 2 toxins-13-00436-t002:** Details of outcome measures.

Outcome	BVA (n = 20)		NSAIDs (n = 20)		Combined (n = 20)		*p-*Value ^+^ (between the 3 Groups)	BVA Versus NSAIDs ^†^	NSAIDs Versus Combined ^†^	BVA Versus Combined ^†^
	Mean (95% CI)	*p-*Value * (within)	Mean (95% CI)	*p-*Value * (within)	Mean (95% CI)	*p*-Value * (within)				
VAS (Bothersomeness)										
Baseline	60.85 (52.68, 69.02)		56.20 (47.60, 64.80)		65.60 (59.65, 71.55)					
Week 2	39.20 (29.95, 48.45)	**<0.0001**	37.15 (28.73, 45.57)	**0.0005**	47.65 (38.96, 56.34)	**<0.0001**	0.3505	−4.96 (−20.47, 10.55)	−3.07 (−17.19, 11.06)	−8.03 (−21.85, 5.79)
Week 3	32.00 (21.78, 42.22)	**<0.0001**	36.75 (27.52, 45.98)	**0.0003**	45.30 (34.84, 55.76)	**0.0002**	**0.0442**	−15.10 (−32.21, 2.02)	1.22 (−14.37, 16.81)	−13.88 (−29.13, 1.37)
Week 4	26.25 (17.70, 34.79)	**<0.0001**	31.65 (21.91, 41.39)	**0.0001**	43.35 (33.62, 53.08)	**<0.0001**	**0.0317**	−11.39 (−28.43, 5.66)	−4.91 (−20.44, 10.62)	**−16.30 (−31.48, −1.11)**
Week 8	26.95 (18.77, 35.13)	**<0.0001**	41.25 (31.37, 51.13)	**0.0012**	40.00 (27.51, 52.49)	**0.0003**	**0.0214**	**−20.93 (−39.37, −2.49)**	9.13 (−7.67, 25.92)	−11.80 (−28.23, 4.63)
VAS (Pain)										
Baseline	56.80 (46.94, 66.66)		50.65 (40.38, 60.92)		64.55 (56.88, 72.22)					
Week 2	36.25 (27.35, 45.15)	**<0.0001**	36.15 (26.86, 45.44)	**0.0007**	47.05 (39.67, 54.43)	**0.0004**	0.3149	−5.44 (−19.32, 8.44)	−2.02 (−14.82, 10.79)	−7.45 (−19.79, 4.88)
Week 3	29.65 (20.19, 39.11)	**<0.0001**	35.50 (26.09, 44.91)	**0.0016**	43.00 (34.07, 51.93)	**0.0002**	**0.0314**	**−15.44 (−30.30, −0.59)**	3.31 (−10.39, 17.01)	−12.13 (−25.33, 1.06)
Week 4	26.60 (17.62, 35.58)	**<0.0001**	31.45 (20.87, 42.03)	**0.0011**	43.25 (34.36, 52.14)	**<0.0001**	**0.0375**	−12.70 (−28.84, 3.43)	−1.71 (−16.61, 13.18)	**−14.42 (−28.76, −0.08)**
Week 8	25.75 (17.65, 33.85)	**<0.0001**	38.85 (27.28, 50.42)	**0.0304**	38.70 (26.19, 51.21)	**0.0002**	0.0864	−17.83 (−37.71, 2.06)	8.31 (−10.04, 26.66)	−9.52 (−27.19, 8.16)
NDI										
Baseline	30.1 (25.0, 35.1)		27.1 (20.8, 33.3)		31.1 (25.5, 36.6)					
Week 2	22.1 (16.3, 27.9)	**0.0004**	23.7 (17.5, 29.8)	0.0628	22.5 (17.3, 27.7)	**0.0007**	**0.0479**	−6.8 (−14.2, 0.6)	5.7 (−0.9, 12.3)	−1.1 (−7.7, 5.4)
Week 3	18.5 (12.6, 24.4)	**<0.0001**	21.7 (15.6, 27.7)	**0.0137**	20.8 (15.9, 25.6)	**0.0003**	0.0733	−7.2 (−14.8, 0.4)	4.7 (−2.1, 11.5)	−2.5 (−9.2, 4.2)
Week 4	17.9 (12.7, 23.0)	**<0.0001**	20.9 (12.8, 28.9)	**0.0196**	22.6 (15.2, 29.9)	**0.0039**	0.0756	−8.6 (−17.9, 0.8)	3.3 (−5.0, 11.7)	−5.2 (−13.5, 3.1)
Week 8	16.7 (10.6, 22.7)	**<0.0001**	24.1 (16.2, 31.9)	0.1706	20.9 (14.4, 27.4)	**0.0008**	**0.0098**	**−11.7 (−21.1, −2.3)**	7.5 (−0.9, 15.9)	−4.2 (−12.5, 4.2)
EQ-5D										
Baseline	0.782 (0.714, 0.850)		0.827 (0.770, 0.884)		0.801 (0.744, 0.858)					
Week 2	0.818 (0.758, 0.879)	0.1810	0.869 (0.824, 0.914)	**0.0084**	0.859 (0.826, 0.892)	**0.0279**	0.6737	−0.006 (−0.078, 0.066)	−0.015 (−0.079, 0.049)	−0.021 (−0.084, 0.042)
Week 3	0.851 (0.790, 0.913)	**0.0036**	0.850 (0.794, 0.906)	0.3553	0.869 (0.838, 0.900)	**0.0079**	0.1624	0.053 (−0.022, 0.128)	−0.048 (−0.114, 0.019)	0.005 (−0.060, 0.071)
Week 4	0.844 (0.784, 0.905)	**0.0036**	0.845 (0.781, 0.909)	0.1000	0.850 (0.806, 0.894)	**0.0420**	0.3985	0.037 (−0.034, 0.107)	−0.027 (−0.089, 0.036)	0.010 (−0.052, 0.072)
Week 8	0.866 (0.817, 0.914)	**0.0011**	0.847 (0.777, 0.916)	0.2041	0.860 (0.817, 0.902)	**0.0169**	0.0588	0.071 (−0.0034, 0.145)	−0.043 (−0.109, 0.023)	0.029 (−0.036, 0.094)
SF-36										
Baseline	63.41 (55.54, 71.27)		66.90 (57.83, 75.97)		62.00 (55.79, 68.21)					
Week 4	68.77 (60.45, 77.09)	**0.0012**	69.82 (59.75, 79.88)	0.1269	65.35 (57.19, 73.50)	0.2276	0.1164	6.86 (−1.41, 15.12)	−2.64 (−10.02, 4.74)	4.22 (−3.11, 11.55)
Week 8	70.93 (62.66, 79.19)	**0.0002**	67.52 (58.13, 76.91)	0.7173	68.42 (60.91, 75.92)	**0.0245**	**0.0182**	**9.44 (1.00, 17.89)**	−6.86 (−14.41, 0.68)	2.58 (−4.91, 10.07)
BDI										
Baseline	10.65 (6.79, 14.51)		9.50 (4.06, 14.94)		10.55 (7.27, 13.83)					
Week 4	5.90 (2.07, 9.73)	**0.0025**	7.80 (2.05, 13.55)	**0.0417**	6.70 (3.26, 10.14)	**0.0040**	0.1496	−3.67 (−8.42, 1.09)	2.42 (−1.80, 6.64)	−1.24 (−5.42, 2.93)
Week 8	4.85 (1.47, 8.23)	**0.0001**	8.20 (2.00, 14.40)	0.1156	6.80 (3.25, 10.35)	**0.0041**	**0.0250**	**−5.07 (−9.61, −0.53)**	0.075 (−0.009, 0.159)	−0.042 (−0.125, 0.042)

BDI, Beck Depression Inventory; BVA, bee venom acupuncture; CI, confidence interval; EQ-5D, EuroQol 5-dimension; NDI, neck disability index; NSAIDs, non-steroidal anti-inflammatory drugs; SF-36, 36-item Short Form Health Survey; VAS, visual analogue scale. * Paired *t*-test, ^+^ Analysis of covariance (covariate: baseline score, age), ^†^ Scheffé adjustment (represented as least squares mean difference and 95% confidence interval). Bold values mean significant differences.

**Table 3 toxins-13-00436-t003:** Comparisons of treatment credibility during the treatment period.

Outcome	BVA (n = 20)		NSAIDs (n = 20)		Combined (n = 20)		*p-*Value ^+^ (between the 3 Groups)
	Mean (95% CI)	*p-*Value * (within)	Mean (95% CI)	*p-*Value * (within)	Mean (95% CI)	*p-*Value * (within)	
Total score							
Baseline	20.20 (18.96, 21.44)		18.80 (17.36, 20.24)		18.30 (16.83, 19.77)		
Week 4	19.75 (18.48, 21.02)	0.5473	17.75 (16.22, 19.28)	**0.0308**	17.95 (16.65, 19.25)	0.6032	0.3108
Expectation of NCNP improvement							
Baseline	5.20 (4.87, 5.53)		4.90 (4.56, 5.24)		4.80 (4.44, 5.16)		
Week 4	5.00 (4.70, 5.30)	0.2967	4.65 (4.21, 5.09)	0.1713	4.60 (4.19, 5.01)	0.3299	0.6682
Willingness to recommend to others							
Baseline	5.15 (4.80, 5.50)		4.65 (4.27, 5.03)		4.65 (4.24, 5.06)		
Week 4	5.00 (4.66, 5.34)	0.481	4.25 (3.73, 4.77)	**0.0421**	4.55 (4.23, 4.87)	0.5409	0.3162
Rationality of treatment							
Baseline	4.95 (4.63, 5.27)		4.70 (4.30, 5.10)		4.50 (4.11, 4.89)		
Week 4	4.85 (4.54, 5.16)	0.5770	4.35 (3.89, 4.81)	**0.0153**	4.40 (3.99, 4.81)	0.6493	0.4537
Effectiveness for alleviating other complaints							
Baseline	4.90 (4.50, 5.30)		4.55 (4.08, 5.02)		4.35 (3.89, 4.81)		
Week 4	4.90 (4.40, 5.40)	0.9999	4.50 (4.06, 4.94)	0.8037	4.40 (4.05, 4.75)	0.8336	0.3064

BVA, bee venom acupuncture; CI, confidence interval; NCNP, non-specific chronic neck pain; NSAIDs, non-steroidal anti-inflammatory drugs. * Paired *t*-test, ^+^ Analysis of covariance (covariate: baseline score, age). Bold values mean significant differences.

**Table 4 toxins-13-00436-t004:** Eligibility criteria.

**Inclusion Criteria**
(1) Age between 18 and 65 years old
(2) Non-specific, uncomplicated, and chronic neck pain for ≥3 months at screening visit
(3) Participants who voluntarily agree to participate and sign written informed consent
**Exclusion Criteria**
(1) Abnormalities on neurological examination
(2) Neck pain with radicular pain
(3) Serious spinal disorders including malignancy, vertebral fracture, spinal infection, and inflammatory spondylitis
(4) Other chronic diseases that were judged to affect the therapeutic outcomes, including cardiovascular disease, diabetic neuropathy, active hepatitis, fibromyalgia, rheumatoid arthritis, dementia, and epilepsy
(5) History of spinal surgery or scheduled spinal surgery during the trial period
(6) Pain induced by a traffic accident
(7) A substantial musculoskeletal problem other than neck pain
(8) Conditions for which the administration of bee venom treatment might not be safe, including clotting disorders, administration of an anticoagulant agent, pregnancy, and seizure disorders
(9) A hypersensitive reaction to previous bee venom treatment, bee stings, or insect bites
(10) A positive reaction observed during a skin hypersensitivity test at screening visit
(11) Diagnosis of severe psychiatric or psychological disorders
(12) Current use of corticosteroids, narcotics, muscle relaxants, or herbal medicines to treat neck pain or use of any medication judged inappropriate by the investigator
(13) Pending lawsuits or receipt of compensation due to neck pain

## Data Availability

The data presented in this study are available on request from the corresponding author.

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
