# Peer review of "Effect of Combined Bee Venom Acupuncture and NSAID Treatment for Non-Specific Chronic Neck Pain: A Randomized, Assessor-Blinded, Pilot Clinical Trial"

_toxins, 2021, doi:10.3390/toxins13070436_

Round 1

Reviewer 1 Report

  • Introduction – In the Introduction, it is necessary to mention the effects of bee venom in the induction of allergies. Check and cite, for example, Clin Exp Allergy. 2019 Jan;49(1):54-67. Also check and cite papers claiming that the BVA can actually be used to desensitize these patients – check and cite, for example, Immunotherapy 2011;3: 229–246 and Pharmacol Ther 2007;115: 246–270.
  • Table 1 – “*Fisher’s exact test, presented as number (%). +Analysis of variance test, pre-sented as mean and 95% confidence interval.” – unclear
  • Chapter 2.1 – “tween groups (P > .05), except age (P < .0001).” – when reporting the p values, mention always the test used and the test statistics.
  • How did you ensure the randomization if the age of each group is so much different? Disclose more details concerning the order of examined patients – I have seen the detailed description of the mechanism of randomization but no proof of how it was applied – for example, an anonymized list of patients with admission dates and assignment could be fine. Also, provide clinical details concerning each such patient, it would be a good opportunity to disclose the raw data.
  • Discussion – “BVA has been used frequently with NSAIDs in Korean clinical medicine” – the journal has an international audience; therefore, it is necessary to comment also on the situation abroad.
  • Venom powder source – you claim that it was obtained from “Yoomil Garden” – however, when googling this company, I found only links to previous papers but not the company itself. Please disclose more details on the product that will allow us to find it and reproduce the performed research.
  • Recent papers are uncited. There are none from 2021.
  • The study was performed long ago. What were the long-term effects of the interventions?
  • Provide a reference for the calculation of the VAS score.
  • Abstract – “tendency to worsen.” – more time points would be needed to claim this – currently, it is a one-point increase only. Therefore, it could be a difference but it also could be some random effect.

Author Response

Comment 1:  

Introduction – In the Introduction, it is necessary to mention the effects of bee venom in the induction of allergies. Check and cite, for example, Clin Exp Allergy. 2019 Jan;49(1):54-67. Also check and cite papers claiming that the BVA can actually be used to desensitize these patients – check and cite, for example, Immunotherapy 2011;3: 229–246 and Pharmacol Ther 2007;115: 246–270.

Response 1:           

Thank you for the careful review of the manuscript. We have added the following sentences to the revised manuscript based on your comments:

“Bee venom induces allergic immune responses such as increased CD16 surface expression [10]. However, its repeated administration can induce therapeutic effects such as anti-nociceptive effects by desensitization [11,12].” (see Page 2, marked in red)

Comment 2: 

Table 1 – “*Fisher’s exact test, presented as number (%). +Analysis of variance test, pre-sented as mean and 95% confidence interval.” – unclear

Response 2:           

Thank you for pointing this out. For clarity, we have revised the text as follows:

“*Statistical analysis was performed using Fisher's exact test, and data are presented as number (%).

+Statistical analysis was performed using analysis of variance, and data are presented as mean (95% confidence interval).” (see Page 4, marked in red)

Comment 3: 

Chapter 2.1 – “tween groups (P > .05), except age (P < .0001).” – when reporting the p values, mention always the test used and the test statistics.

Response 3:           

We have added the test used with P values in the revised manuscript as follows:

“between the groups (P > .05; Fisher’s exact test (sex, smoke, and drink) and analysis of variance (ANOVA) (height, weight, body mass index, and vital signs)), except age (P < .0001; ANOVA).” (see Page 3, marked in red)

Comment 4: 

How did you ensure the randomization if the age of each group is so much different? Disclose more details concerning the order of examined patients – I have seen the detailed description of the mechanism of randomization but no proof of how it was applied for example, an anonymized list of patients with admission dates and assignment could be fine. Also, provide clinical details concerning each such patient, it would be a good opportunity to disclose the raw data.

Response 4:           

Thank you for these important suggestions. Because pre-generated random numbers were assigned according to the order of participant enrollment, age did not affect randomization. However, when analyzing the results, because the age of the NSAIDs group was significantly lower than that of the other groups, age was used as a covariate for the outcome analysis to minimize potential bias (described in Page 3). Additionally, according to your comment, we have added an anonymized list of study participants with the screening, enrollment, and visit days and assigned groups, as Supplementary Materials (Table S3).

Comment 5: 

Discussion – “BVA has been used frequently with NSAIDs in Korean clinical medicine” – the journal has an international audience; therefore, it is necessary to comment also on the situation abroad.

Response 5:           

We have modified the sentence as follows to introduce studies in which BVA was used for pain control in East Asian countries in the revised manuscript:

“In East Asia, BVA combined with conventional medication, including NSAIDs, has been used to treat cancer-related pain [28] and low back pain [29], and especially, its clinical effect for NCNP has been well studied [13,19].” (see Page 9, marked in red)

Comment 6: 

Venom powder source – you claim that it was obtained from “Yoomil Garden” – however, when googling this company, I found only links to previous papers but not the company itself. Please disclose more details on the product that will allow us to find it and reproduce the performed research.

Response 6:           

Thank you for the valuable comments. In our study, bee venom was prepared according to the guidelines for the in-hospital preparation of injections in accordance with the inspection standards of the Korean Ministry of Food and Drug Safety. We have added detailed description of the method for preparing bee venom to the revised manuscript as follows:

“A hundred milliliters of normal saline was added to 5 mg of dried bee venom powder and dissolved well; after filtering with a 0.22 μg filter, 5 ml of each was filled in a sterilized vial. After confirming the suitability by conducting an environmental culture microorganism test, it was refrigerated at 4-6°C.” (see Page 12, marked in red)

In addition, information related to “Yoomil Garden” can be found at the following URL: https://g.co/kgs/GW6DCX

Comment 7: 

Recent papers are uncited. There are none from 2021.

Response 7:           

Based on your comment, we have now added several recent studies related to BVA, published in 2021, as references in the revised manuscript (References 14, 15, and 28, marked in red).

Comment 8: 

The study was performed long ago. What were the long-term effects of the interventions?

Response 8:           

Thank you for the comment. In our study, follow-up evaluation was performed only in Weeks 4 and 8 after 3 weeks of treatment, and we found that BVA had a lasting effect for 5 weeks after the end of the treatment. However, subsequent long-term effects could not be confirmed in our study. We have mentioned that additional long-term effects studies of combined treatment of BVA and NSAIDs are needed through full-scale randomized controlled trials and registry-based studies in the Discussion section of the revised manuscript.

“Therefore, future studies should examine the long-term lasting effect of the combined treatment of BVA and NSAIDs through longer follow-up duration with full-scale randomized controlled trials and registry-based studies.” (see Page 9, marked in red).

Comment 9: 

Provide a reference for the calculation of the VAS score.

Response 9:           

We have added references for the calculation of the VAS score in the revised manuscript (References 40 and 41).

Comment 10: 

Abstract – “tendency to worsen.” – more time points would be needed to claim this – currently, it is a one-point increase only. Therefore, it could be a difference but it also could be some random effect.

Response 10:         

Based on your comment, we have deleted the text “tendency to worsen” in the Abstract of the revised manuscript.

Reviewer 2 Report

Presented paper is valuable both from research side (scientific outcome, methodology, statistics) and social impact side (raise of health and welfare feeling). 

Author Response

Thank you for the careful review of the manuscript.

Reviewer 3 Report

Chronic pain is a major health and economic burden, with collateral health problems arising from the use of drugs such as opoids, thus investigation of alternative methods for pain relief, including combinations which may work by  complementary pathways is worthwhile.  Bee venom acupuncture has been studied for some time and its potential and side effects have been pointed out (Toxicon . 2018 Nov;154:74-78.  doi: 10.1016/j.toxicon.2018.09.013. Epub 2018 Sep 28.  

The present study appears to have been carefully designed and monitored. The 3 groups were reasonable well matched, the average age of the combined treatment group being in between that of the single treatments. I'm unable to comment with any authority on the statistical analyses. However I think that the abstract should be more specific in stating p values of confidence limits for the differences that are considered to be significant. Also, a qualification about variability of bee venoms and a warning given about the potential for anaphylactic reactions; in case  someone wants to follow up.

Author Response

Thank you for careful review of the manuscript. According to your advice, the P value for the results has been added to the Abstract and the precautions for BVA have been added to the Discussion section as follows:

“In addition, BVA, if not used properly, can cause systemic adverse reactions, such as anaphylactic reactions [36]; therefore, it should be performed by a qualified professional who has completed relevant formal training.” (see Page 10, marked in red).

Round 2

Reviewer 1 Report

Thank you for the performed changes, the manuscript looks fine now. However, one issue remains to be solved, and it is related to the reports of the statistical evaluations. In your rebuttal letter, you claim that:

Response 3:

We have added the test used with P values in the revised manuscript as follows:

“between the groups (P > .05; Fisher’s exact test (sex, smoke, and drink) and analysis of variance (ANOVA) (height, weight, body mass index, and vital signs)), except age (P < .0001; ANOVA).” (see Page 3, marked in red)

However, the report needs to be more detailed. For example, for ANOVA, it is necessary to 1) state the type of ANOVA (one-way?), 2) to state the F values (in addition to the P values), and 3) when reporting that there were differences between individual groups ("except age"), then it is necessary to disclose the type of the post-test performed. 4) Also, when reporting ANOVA, make sure that you tested for data distribution normality and variance equality. 5) And the last point, the use of ANOVA is not mentioned in the Statistics section, there is only RM-ANOVA mentioned. Please make sure that all the methods are mentioned in the Methods chapter.

Author Response

1) We used one-way ANOVA and added this to the manuscript.

2) We added the F value to Table 1 (see Pages 3-4, marked in red).

3) We added the post-hoc tests used as follows to the revised manuscript.

“There were no significant differences in the baseline demographic characters between the groups (P > .05; Fisher’s exact test (sex, smoke, and drink) and one-way analysis of variance (ANOVA) (height, weight, body mass index, and vital signs)), except age (P < .0001; one-way ANOVA). Post-hoc analysis, according to Scheffe's method, showed that the age of the NSAIDs group was significantly lower than the other groups; therefore, age was used as a covariate for the outcome analysis to minimize potential bias.” (see Page 3, marked in red).

4) Kolmogorov-Smirniv test and Shapiro-Wilk test were used to assess the normality of distribution. Also, Levene's test was used to assess the equality of variances. We added this information to the revised manuscript.

5) We added all statistical analysis methods used to the revised manuscript as follows.

“4.7. Statistical Analysis

All statistical analyses were performed using SAS Version 9.4 (SAS institute, Inc., Cary, NC, USA) according to the intention-to-treatment analysis principle by a statistician independent of treatment procedure and evaluation. Normality of data distribution was confirmed by Kolmogorov-Smirniv test and Shapiro-Wilk test, and equal variance was confirmed by Levene's test. For baseline characteristic of participants, one-way ANOVA was performed for continuous variables, and Fisher's exact test was performed for categorical variables. A two-sided test with a significance level of 0.05 was performed using an analysis of covariance (ANCOVA) with baseline as the covariate and each group as the fixed factor to test the differences between the three groups. Next, post-hoc Scheffe adjustment was performed to compare the differences between two groups. A paired t-test was used to analyze the changes in the results before and after treatment within the groups. RM-ANOVA was performed to explore interactions between time and treatment group. Missing values were corrected by applying the last observation carried forward method. Interim analysis was not performed.“ (see Page 14, marked in red)